# Smoothed analysis of the low-rank approach for smooth semidefinite programs

**Thomas Pumir**[*]
ORFE Department
Princeton University
tpumir@princeton.edu

**Samy Jelassi**[*]
ORFE Department
Princeton University
sjelassi@princeton.edu

**Nicolas Boumal**
Department of Mathematics
Princeton University
nboumal@math.princeton.edu

## Abstract

We consider semidefinite programs (SDPs) of size $n$ with equality constraints. In order to overcome scalability issues, Burer and Monteiro proposed a factorized approach based on optimizing over a matrix $Y$ of size $n \times k$ such that $X = YY^*$ is the SDP variable. The advantages of such formulation are twofold: the dimension of the optimization variable is reduced, and positive semidefiniteness is naturally enforced. However, optimization in $Y$ is non-convex. In prior work, it has been shown that, when the constraints on the factorized variable regularly define a smooth manifold, provided $k$ is large enough, for almost all cost matrices, all second-order stationary points (SOSPs) are optimal. Importantly, in practice, one can only compute points which approximately satisfy necessary optimality conditions, leading to the question: are such points also approximately optimal? To answer it, under similar assumptions, we use smoothed analysis to show that approximate SOSPs for a randomly perturbed objective function are approximate global optima, with $k$ scaling like the square root of the number of constraints (up to log factors). Moreover, we bound the optimality gap at the approximate solution of the perturbed problem with respect to the original problem. We particularize our results to an SDP relaxation of phase retrieval.

## 1 Introduction

We consider semidefinite programs (SDP) over $\mathbb{K} = \mathbb{R}$ or $\mathbb{C}$ of the form:

$$\min_{X \in \mathbb{S}^{n \times n}} \quad \langle C, X \rangle$$
$$\text{subject to} \quad \mathcal{A}(X) = b, \tag{SDP}$$
$$X \succeq 0,$$

with $\langle A, B \rangle = \operatorname{Re}[\operatorname{Tr}(A^*B)]$ the Frobenius inner product ($A^*$ is the conjugate-transpose of $A$), $\mathbb{S}^{n \times n}$ the set of self-adjoint matrices of size $n$ (real symmetric for $\mathbb{R}$, or Hermitian for $\mathbb{C}$), $C \in \mathbb{S}^{n \times n}$ the cost matrix, and $\mathcal{A} \colon \mathbb{S}^{n \times n} \to \mathbb{R}^m$ a linear operator capturing $m$ equality constraints with right hand side $b \in \mathbb{R}^m$: for each $i$, $\mathcal{A}(X)_i = \langle A_i, X \rangle = b_i$ for given matrices $A_1, \ldots, A_m \in \mathbb{S}^{n \times n}$. The optimization variable $X$ is positive semidefinite. We let $\mathcal{C}$ be the feasible set of (SDP):

$$\mathcal{C} = \left\{ X \in \mathbb{S}^{n \times n} : \mathcal{A}(X) = b \text{ and } X \succeq 0 \right\}. \tag{1}$$

---

[*]Equal contribution

Large-scale SDPs have been proposed for machine learning applications including matrix completion [Candès and Recht, 2009], community detection [Abbé, 2018] and kernel learning [Lanckriet et al., 2004] for $\mathbb{K} = \mathbb{R}$, and in angular synchronization [Singer, 2011] and phase retrieval [Wald-spurger et al., 2015] for $\mathbb{K} = \mathbb{C}$. Unfortunately, traditional methods to solve (SDP) do not scale (due to memory and computational requirements), hence the need for alternatives.

In order to address such scalability issues, Burer and Monteiro [2003, 2005] restrict the search to the set of matrices of rank at most $k$ by factorizing $X$ as $X = YY^*$, with $Y \in \mathbb{K}^{n \times k}$. It has been shown that if the search space $\mathcal{C}$ (1) is compact, then (SDP) admits a global optimum of rank at most $r$, where $\dim \mathbb{S}^{r \times r} \leq m$ [Barvinok, 1995, Pataki, 1998], with $\dim \mathbb{S}^{r \times r} = \frac{r(r+1)}{2}$ for $\mathbb{K} = \mathbb{R}$ and $\dim \mathbb{S}^{r \times r} = r^2$ for $\mathbb{K} = \mathbb{C}$. In other words, restricting $\mathcal{C}$ to the space of matrices with rank at most $k$ with $\dim \mathbb{S}^{k \times k} > m$ does not change the optimal value. This factorization leads to a quadratically constrained quadratic program:

$$\min_{Y \in \mathbb{K}^{n \times k}} \quad \langle C, YY^* \rangle$$
$$\text{subject to} \quad \mathcal{A}(YY^*) = b. \tag{P}$$

Although (P) is in general non-convex because its feasible set

$$\mathcal{M} = \mathcal{M}_k = \left\{ Y \in \mathbb{K}^{n \times k} : \mathcal{A}(YY^*) = b \right\} \tag{2}$$

is non-convex, considering (P) instead of the original SDP presents significant advantages: the number of variables is reduced from $O(n^2)$ to $O(nk)$, and the positive semidefiniteness of the matrix is naturally enforced. Solving (P) using local optimization methods is known as the Burer–Monteiro method and yields good results in practice: Kulis et al. [2007] underlined the practical success of such low-rank approaches in particular for maximum variance unfolding and for k-means clustering (see also [Carson et al., 2017]). Their approach is significantly faster and more scalable. However, the non-convexity of (P) means further analysis is needed to determine whether it can be solved to global optimality reliably.

For $\mathbb{K} = \mathbb{R}$, in the case where $\mathcal{M}$ is a compact, smooth manifold (see Assumption 1 below for a precise condition), it has been shown recently that, up to a zero-measure set of cost matrices, second-order stationary points (SOSPs) of (P) are globally optimal provided $\dim \mathbb{S}^{k \times k} > m$ [Boumal et al., 2016, 2018b]. Algorithms such as the Riemannian trust-regions method (RTR) converge globally to SOSPs, but unfortunately they can only guarantee *approximate* satisfaction of second-order optimality conditions in a finite number of iterations [Boumal et al., 2018a].

The aforementioned papers close with a question, crucial in practice: when is it the case that *approximate* SOSPs, which we now call ASOSPs, are approximately optimal? Building on recent proof techniques by Bhojanapalli et al. [2018], we provide some answers here.

## Contributions

This paper formulates approximate global optimality conditions holding for (P) and, consequently, for (SDP). Our results rely on the following core assumption as set in [Boumal et al., 2016].

**Assumption 1** (Smooth manifold)**.** *For all values of $k$ up to $n$ such that $\mathcal{M}_k$ is non-empty, the constraints on* (P) *defined by $A_1, \ldots, A_m \in \mathbb{S}^{n \times n}$ and $b \in \mathbb{R}^m$ satisfy at least one of the following:*

1. *$\{A_1 Y, \ldots, A_m Y\}$ are linearly independent in $\mathbb{K}^{n \times k}$ for all $Y \in \mathcal{M}_k$; or*

2. *$\{A_1 Y, \ldots, A_m Y\}$ span a subspace of constant dimension in $\mathbb{K}^{n \times k}$ for all $Y$ in an open neighborhood of $\mathcal{M}_k$ in $\mathbb{K}^{n \times k}$.*

In [Boumal et al., 2018b], it is shown that (a) if the assumption above is verified for $k = n$, then it automatically holds for all values of $k \leq n$ such that $\mathcal{M}_k$ is non-empty; and (b) for those values of $k$, the dimension of the subspace spanned by $\{A_1 Y, \ldots, A_m Y\}$ is independent of $k$: we call it $m'$.

When Assumption 1 holds, we refer to problems of the form (SDP) as *smooth* SDPs because $\mathcal{M}$ is then a smooth manifold. Examples of smooth SDPs for $\mathbb{K} = \mathbb{R}$ are given in [Boumal et al., 2018b]. For $\mathbb{K} = \mathbb{C}$, we detail an example in Section 4. Our main theorem is a smooth analysis result (cf. Theorem 3.1 for a more formal statement). An ASOSP is an *approximate* SOSP (a precise definition follows.)

**Theorem 1.1** (Informal)**.** *Let Assumption 1 hold and assume $\mathcal{C}$ is compact. Randomly perturb the cost matrix $C$. With high probability, if $k = \tilde{\Omega}(\sqrt{m})$, any ASOSP $Y \in \mathbb{K}^{n \times k}$ for (P) is an approximate global optimum, and $X = YY^*$ is an approximate global optimum for (SDP) (with the perturbed $C$.)*

The high probability proviso is with respect to the perturbation only: if the perturbation is "good", then all ASOSPs are as described in the statement. If $\mathcal{C}$ is compact, then so is $\mathcal{M}$ and known algorithms for optimization on manifolds produce an ASOSP in finite time (with explicit bounds). Theorem 1.1 ensures that, for $k$ large enough and for any cost matrix $C$, with high probability upon a random perturbation of $C$, such algorithms produce an approximate global optimum of (P).

Theorem 1.1 is a corollary of two intermediate arguments, developed in Lemmas 3.1 and 3.2:

1. Probabilistic argument (Lemma 3.1): By perturbing the cost matrix in the objective function of (P) with a Gaussian Wigner matrix, with high probability, any approximate first-order stationary point $Y$ of the perturbed problem (P) is almost column-rank deficient.

2. Deterministic argument (Lemma 3.2): If an approximate second-order stationary point $Y$ for (P) is also almost column-rank deficient, then it is an approximate global optimum and $X = YY^*$ is an approximate global optimum for (SDP).

The first argument is motivated by *smoothed analysis* [Spielman and Teng, 2004] and draws heavily on a recent paper by Bhojanapalli et al. [2018]. The latter work introduces smoothed analysis to analyze the performance of the Burer–Monteiro factorization, but it analyzes a quadratically penalized version of the SDP: its solutions do not satisfy constraints exactly. This affords more generality, but, for the special class of smooth SDPs, the present work has the advantage of analyzing an exact formulation. The second argument is a smoothed extension of well-known on-off results [Burer and Monteiro, 2003, 2005, Journee et al., 2010]. Implications of this theorem for a particular SDP are derived in Section 4, with applications to phase retrieval and angular synchronization.

Thus, for smooth SDPs, our results improve upon [Bhojanapalli et al., 2018] in that we address exact-feasibility formulations of the SDP. Our results also improve upon [Boumal et al., 2016] by providing approximate optimality results for approximate second-order points with relaxation rank $k$ scaling only as $\tilde{\Omega}(\sqrt{m})$, whereas the latter reference establishes such results only for $k = n + 1$. Finally, we aim for more generality by covering both real and complex SDPs, and we illustrate the relevance of complex SDPs in Section 4.

**Related work**

A number of recent works focus on large-scale SDP solvers. Among the direct approaches (which proceed in the convex domain directly), Hazan [2008] introduced a Frank–Wolfe type method for a restricted class of SDPs. Here, the key is that each iteration increases the rank of the solution only by one, so that if only a few iterations are required to reach satisfactory accuracy, then only low dimensional objects need to be manipulated. This line of work was later improved by Laue [2012], Garber [2016] and Garber and Hazan [2016] through hybrid methods. Still, if high accuracy solutions are desired, a large number of iterations will be required, eventually leading to large-rank iterates. In order to overcome such issue, Yurtsever et al. [2017] recently proposed to combine conditional gradient and sketching techniques in order to maintain a low rank representation of the iterates.

Among the low-rank approaches, our work is closest to (and indeed largely builds upon) recent results of Bhojanapalli et al. [2018]. For the real case, they consider a penalized version of problem (SDP) (which we here refer to as (P-SDP)) and its related penalized Burer–Monteiro formulation, here called (P-P). With high probability upon random perturbation of the cost matrix, they show approximate global optimality of ASOSPs for (P-P), assuming $k$ grows with $\sqrt{m}$ and either the SDP is compact or its cost matrix is positive definite. Given that there is a zero-measure set of SDPs where SOSPs may be suboptimal, there can be a small-measure set of SDPs where ASOSPs are not approximately optimal [Bhojanapalli et al., 2018]. In this context, the authors resort to smoothed analysis, in the same way that we do here. One drawback of that work is that the final result does not hold for the original SDP, but for a non-equivalent penalized version of it. This is one of the points we improve here, by focusing on smooth SDPs as defined in [Boumal et al., 2016].

**Notation**

We use $\mathbb{K}$ to refer to $\mathbb{R}$ or $\mathbb{C}$ when results hold for both fields. For matrices $A$, $B$ of same size, we use the inner product $\langle A, B \rangle = \mathrm{Re}[\mathrm{Tr}(A^*B)]$, which reduces to $\langle A, B \rangle = \mathrm{Tr}(A^T B)$ in the real case. The associated Frobenius norm is defined as $\|A\| = \sqrt{\langle A, A \rangle}$. For a linear map $f$ between matrix spaces, this yields a subordinate operator norm as $\|f\|_{\mathrm{op}} = \sup_{A \neq 0} \frac{\|f(A)\|}{\|A\|}$. The set of self-adjoint matrices of size $n$ over $\mathbb{K}$, $\mathbb{S}^{n \times n}$, is the set of symmetric matrices for $\mathbb{K} = \mathbb{R}$ or the set of Hermitian matrices for $\mathbb{K} = \mathbb{C}$. We also write $\mathbb{H}^{n \times n}$ to denote $\mathbb{S}^{n \times n}$ for $\mathbb{K} = \mathbb{C}$. A self-adjoint matrix $X$ is positive semidefinite ($X \succeq 0$) if and only if $u^* X u \geq 0$ for all $u \in \mathbb{K}^n$. Furthermore, $I$ is the identity operator and $I_n$ is the identity matrix of size $n$. The integer $m'$ is defined after Assumption 1.

## 2  Geometric framework and near-optimality conditions

In this section, we present properties of the smooth geometry of (P) and approximate global optimality conditions for this problem. In covering these preliminaries, we largely parallel developments in [Boumal et al., 2016]. As argued in that reference, Assumption 1 implies that the search space $\mathcal{M}$ of (P) is a submanifold in $\mathbb{K}^{n \times k}$ of codimension $m'$. We can associate tangent spaces to a submanifold. Intuitively, the tangent space $\mathrm{T}_Y \mathcal{M}$ to the submanifold $\mathcal{M}$ at a point $Y \in \mathcal{M}$ is a subspace that best approximates $\mathcal{M}$ around $Y$, when the subspace origin is translated to $Y$. It is obtained by linearizing the equality constraints.

**Lemma 2.1** (Boumal et al. [2018b, Lemma 2.1])**.** *Under Assumption 1, the tangent space at $Y$ to $\mathcal{M}$ (2), denoted by $\mathrm{T}_Y \mathcal{M}$, is:*

$$\mathrm{T}_Y \mathcal{M} = \left\{ \dot{Y} \in \mathbb{K}^{n \times k} \colon \mathcal{A}(\dot{Y} Y^* + Y \dot{Y}^*) = 0 \right\}$$
$$= \left\{ \dot{Y} \in \mathbb{K}^{n \times k} \colon \langle A_i Y, \dot{Y} \rangle = 0 \ for \ i = 1, \ldots, m \right\}. \tag{3}$$

By equipping each tangent space with a restriction of the inner product $\langle \cdot, \cdot \rangle$, we turn $\mathcal{M}$ into a Riemannian submanifold of $\mathbb{K}^{n \times k}$. We also introduce the orthogonal projector $\mathrm{Proj}_Y \colon \mathbb{K}^{n \times k} \to \mathrm{T}_Y \mathcal{M}$ which, given a matrix $Z \in \mathbb{K}^{n \times k}$, projects it to the tangent space $\mathrm{T}_Y \mathcal{M}$:

$$\mathrm{Proj}_Y Z := \underset{\dot{Y} \in \mathrm{T}_Y \mathcal{M}}{\mathrm{argmin}} \ \|\dot{Y} - Z\|. \tag{4}$$

This projector will be useful to phrase optimality conditions. It is characterized as follows.

**Lemma 2.2** (Boumal et al. [2018b, Lemma 2.2])**.** *Under Assumption 1, the orthogonal projector admits the closed form*

$$\mathrm{Proj}_Y Z = Z - \mathcal{A}^* \left( G^\dagger \mathcal{A}(ZY^*) \right) Y,$$

*where $\mathcal{A}^* \colon \mathbb{R}^m \to \mathbb{S}^{n \times n}$ is the adjoint of $\mathcal{A}$, $G$ is a Gram matrix defined by $G_{ij} = \langle A_i Y, A_j Y \rangle$ (it is a function of $Y$), and $G^\dagger$ denotes the Moore–Penrose pseudo-inverse of $G$ (differentiable in $Y$).*

(See a proof in Appendix A.) To properly state the approximate first- and second-order necessary optimality conditions for (P), we further need the notions of *Riemannian gradient* and *Riemannian Hessian* on the manifold $\mathcal{M}$. We recall that (P) minimizes the function $g$, defined by

$$g(Y) = \langle CY, Y \rangle, \tag{5}$$

on the manifold $\mathcal{M}$. The Riemannian gradient of $g$ at $Y$, $\mathrm{grad}\, g(Y)$, is the unique tangent vector at $Y$ such that, for all tangent $\dot{Y}$, $\langle \mathrm{grad}\, g(Y), \dot{Y} \rangle = \langle \nabla g(Y), \dot{Y} \rangle$, with $\nabla g(Y) = 2CY$ the Euclidean (classical) gradient of $g$ evaluated at $Y$. Intuitively, $\mathrm{grad}\, g(Y)$ is the tangent vector at $Y$ that points in the steepest ascent direction for $g$ as seen from the manifold's perspective. A classical result states that, for Riemannian submanifolds, the Riemannian gradient is given by the projection of the classical gradient to the tangent space [Absil et al., 2008, eq. (3.37)]:

$$\mathrm{grad}\, g(Y) = \mathrm{Proj}_Y(\nabla g(Y)) = 2 \left( C - \mathcal{A}^* \left( G^\dagger \mathcal{A}(CYY^*) \right) \right) Y. \tag{6}$$

This leads us to define the matrix $S \in \mathbb{S}^{n \times n}$ which plays a key role to guarantee approximate global optimality for problem (P), as discussed in Section 3:

$$S = S(Y) = C - \mathcal{A}^*(\mu) = C - \sum_{i=1}^{m} \mu_i A_i, \tag{7}$$

where $\mu = \mu(Y) = G^\dagger \mathcal{A}(CYY^*)$. We can write the Riemannian gradient of $g$ evaluated at $Y$ as

$$\operatorname{grad} g(Y) = 2SY. \tag{8}$$

The Riemannian gradient enables us to define an approximate first-order necessary optimality condition below. To define the approximate second-order necessary optimality condition, we need to introduce the notion of Riemannian Hessian. The Riemannian Hessian of $g$ at $Y$ is a self-adjoint operator on the tangent space at $Y$ obtained as the projection of the derivative of the Riemannian gradient vector field [Absil et al., 2008, eq. (5.15)]. Boumal et al. [2018b] give a closed form expression for the Riemannian Hessian of $g$ at $Y$:

$$\operatorname{Hess} g(Y)[\dot{Y}] = 2 \cdot \operatorname{Proj}_Y(S\dot{Y}). \tag{9}$$

We can now formally define the approximate necessary optimality conditions for problem (P).

**Definition 2.1** ($\varepsilon_g$-FOSP). $Y \in \mathcal{M}$ is an $\varepsilon_g$–first-order stationary point for (P) if the norm of the Riemannian gradient of $g$ at $Y$ almost vanishes, specifically,

$$\|\operatorname{grad} g(Y)\| = \|2SY\| \leq \varepsilon_g,$$

where $S$ is defined as in equation (7).

**Definition 2.2** (($\varepsilon_g, \varepsilon_H$)-SOSP). $Y \in \mathcal{M}$ is an ($\varepsilon_g, \varepsilon_H$)–second-order stationary point for (P) if it is an $\varepsilon_g$–first-order stationary point and the Riemannian Hessian of $g$ at $Y$ is almost positive semidefinite, specifically,

$$\forall \dot{Y} \in \mathrm{T}_Y\mathcal{M}, \qquad \frac{1}{2}\left\langle \dot{Y}, \operatorname{Hess} g(Y)[\dot{Y}] \right\rangle = \langle \dot{Y}, S\dot{Y} \rangle \geq -\varepsilon_H \|\dot{Y}\|^2.$$

From these definitions, it is clear that $S$ encapsulates the approximate optimality conditions of problem (P).

## 3  Approximate second-order points and smoothed analysis

We state our main results formally in this section. As announced, following [Bhojanapalli et al., 2018], we resort to smoothed analysis [Spielman and Teng, 2004]. To this end, we consider perturbations of the cost matrix $C$ of (SDP) by a *Gaussian Wigner matrix*. Intuitively, smoothed analysis tells us how large the variance of the perturbation should be in order to obtain a new SDP which, with high probability, is sufficiently distant from any pathological case. We start by formally defining the notion of Gaussian Wigner matrix, following [Ben Arous and Guionnet, 2010].

**Definition 3.1** (Gaussian Wigner matrix). *The random matrix $W = W^*$ in $\mathbb{S}^{n \times n}$ is a Gaussian Wigner matrix with variance $\sigma_W^2$ if its entries on and above the diagonal are independent, zero-mean Gaussian variables (real or complex depending on context) with variance $\sigma_W^2$.*

Besides Assumption 1, another important assumption for our results is that the search space $\mathcal{C}$ (1) of (SDP) is compact. In that scenario, there exists a finite constant $R$ such that

$$\forall X \in \mathcal{C}, \quad \operatorname{Tr}(X) \leq R. \tag{10}$$

Thus, for all $Y \in \mathcal{M}$, $\|Y\|^2 = \operatorname{Tr}(YY^*) \leq R$. Another consequence of compactness of $\mathcal{C}$ is that the operator $\mathcal{A}^* \circ G^\dagger \circ \mathcal{A}$ is uniformly bounded, that is, there exists a finite constant $K$ such that

$$\forall Y \in \mathcal{M}, \quad \|\mathcal{A}^* \circ G^\dagger \circ \mathcal{A}\|_{\mathrm{op}} \leq K, \tag{11}$$

where $G^\dagger$ is a continuous function of $Y$ as in Lemma 2.2. We give explicit expressions for the constants $R$ and $K$ for the case of phase retrieval in Section 4.

We now state the main theorem, whose proof is in Appendix E.

**Theorem 3.1.** *Let Assumption 1 hold for (SDP) with cost matrix $C \in \mathbb{S}^{n \times n}$ and $m$ constraints. Assume $\mathcal{C}$ (1) is compact, and let $R$ and $K$ be as in (10) and (11). Let $W$ be a Gaussian Wigner matrix with variance $\sigma_W^2$ and let $\delta \in (0, 1)$ be any tolerance. Define $\kappa$ as:*

$$\kappa = \kappa(R, K, C, n, \sigma_W) = RK \left( \|C\|_{\mathrm{op}} + 3\sigma_W\sqrt{n} \right). \tag{12}$$

*There exists a universal constant $c_0$ such that, if the rank $k$ for the low-rank problem* (P) *satisfies*

$$k \geq 3\left[\log(n) + \sqrt{\log(1/\delta)} + \sqrt{m \cdot \log\left(1 + \frac{6\kappa\sqrt{c_0 n}}{\sigma_W}\right)}\right], \tag{13}$$

*then, with probability at least $1 - \delta - e^{-\frac{n}{2}}$ on the random matrix $W$, any $(\varepsilon_g, \varepsilon_H)$-SOSP $Y \in \mathbb{K}^{n \times k}$ of* (P) *with perturbed cost matrix $C + W$ has bounded optimality gap:*

$$0 \leq g(Y) - f^{\star} \leq (\varepsilon_H + \varepsilon_g^2 \eta)R + \frac{\varepsilon_g}{2}\sqrt{R}, \tag{14}$$

*with $g$ the cost function of* (P)*, $f^{\star}$ the optimal value of* (SDP) *(both perturbed), and*

$$\eta = \eta(R, K, C, n, m, \sigma_W) = \frac{c_0 n K(2 + KR)^2\left(\|C\|_{\mathrm{op}} + 3\sigma_W\sqrt{n}\right)}{9m\sigma_W^2 \log\left(1 + \frac{6\kappa\sqrt{c_0 n}}{\sigma_W}\right)}. \tag{15}$$

This result indicates that, as long as the rank $k$ is on the order of $\sqrt{m}$ (up to logarithmic factors), the optimality gap in the *perturbed* problem is small if a sufficiently good *approximate* second-order point is computed. Since (SDP) may admit a unique solution of rank as large as $\Theta(\sqrt{m})$ (see for example [Laurent and Poljak, 1996, Thm. 3.1(ii)] for the Max-Cut SDP), we conclude that the scaling of $k$ with respect to $m$ in Theorem 3.1 is essentially optimal.

There is an incentive to pick $\sigma_W$ small, since the optimality gap is phrased in terms of the perturbed problem. As expected though, taking $\sigma_W$ small comes at a price. Specifically, the required rank $k$ scales with $\sqrt{\log(1/\sigma_W)}$, so that a smaller $\sigma_W$ may require $k$ to be a larger multiple of $\sqrt{m}$. Furthermore, the optimality gap is bounded in terms of $\eta$ with a dependence in $\varepsilon_g^2/\sigma_W^2$; this may force us to compute more accurate approximate second-order points (smaller $\varepsilon_g$) for a similar guarantee when $\sigma_W$ is smaller: see also Corollary 3.1 below.

As announced, the theorem rests on two arguments which we now present—a probabilistic one, and a deterministic one:

1. Probabilistic argument: In the smoothed analysis framework, we show, for $k$ large enough, that $\varepsilon_g$-FOSPs of (P) have their smallest singular value near zero, with high probability upon perturbation of $C$. This implies that such points are almost column-rank deficient.

2. Deterministic argument: If $Y$ is an $(\varepsilon_g, \varepsilon_H)$-SOSP of (P) and it is almost column-rank deficient, then the matrix $S(Y)$ defined in equation (7) is almost positive semidefinite. From there, we can derive a bound on the optimality gap.

Formal statements for both follow, building on the notation in Theorem 3.1. Proofs are in Appendices C and D, with supporting lemmas in Appendix B.

**Lemma 3.1.** *Let Assumption 1 hold for* (SDP)*. Assume $\mathcal{C}$ (1) is compact. Let $W$ be a Gaussian Wigner matrix with variance $\sigma_W^2$ and let $\delta \in (0, 1)$ be any tolerance. There exists a universal constant $c_0$ such that, if the rank $k$ for the low-rank problem* (P) *is lower bounded as in* (13)*, then, with probability at least $1 - \delta - e^{-\frac{n}{2}}$ on the random matrix $W$, we have*

$$\|W\|_{\mathrm{op}} \leq 3\sigma_W\sqrt{n},$$

*and furthermore: any $\varepsilon_g$-FOSP $Y \in \mathbb{K}^{n \times k}$ of* (P) *with perturbed cost matrix $C + W$ satisfies*

$$\sigma_k(Y) \leq \frac{\varepsilon_g}{\sigma_W}\frac{\sqrt{c_0 n}}{k},$$

*where $\sigma_k(Y)$ is the kth singular value of the matrix $Y$.*

**Lemma 3.2.** *Let Assumption 1 hold for* (SDP) *with cost matrix $C$. Assume $\mathcal{C}$ is compact. Let $Y \in \mathbb{K}^{n \times k}$ be an $(\varepsilon_g, \varepsilon_H)$-SOSP of* (P) *(for any $k$). Then, the smallest eigenvalue of $S = S(Y)$ (7) is bounded below as*

$$\lambda_{\min}(S) \geq -\varepsilon_H - \zeta\|C\|_{\mathrm{op}} \cdot \sigma_k^2(Y),$$

*where $\zeta = K(2 + KR)^2$ with $R, K$ as in* (10) *and* (11)*, and $\sigma_k(Y)$ is the kth singular value of $Y$ (it is zero if $k > n$). This holds deterministically for any cost matrix $C$.*

Combining the two above lemmas, the key step in the proof of Theorem 3.1 is to deduce a bound on the optimality gap from a bound on the smallest eigenvalue of $S$: see Appendix E.

We have shown in Theorem 3.1 that a perturbed version of (P) can be approximately solved to global optimality, with high probability on the perturbation. In the corollary below, we further bound the optimality gap at the approximate solution of the perturbed problem with respect to the original, unperturbed problem. The proof is in Appendix F.

**Corollary 3.1.** *Assume $\mathcal{C}$ is compact and let $R$ be as defined in* (10). *Let $X \in \mathcal{C}$ be an approximate solution for* (SDP) *with perturbed cost matrix $C + W$, so that the optimality gap in the perturbed problem is bounded by $\varepsilon_f$. Let $f^\star$ denote the optimal value of the unperturbed problem* (SDP), *with cost matrix $C$. Then, the optimality gap for $X$ with respect to the unperturbed problem is bounded as:*

$$0 \leq \langle C, X \rangle - f^\star \leq \varepsilon_f + 2\|W\|_{\mathrm{op}}R.$$

*Under the conditions of Theorem 3.1, with the prescribed probability, $\varepsilon_f$ and $\|W\|_{\mathrm{op}}$ can be bounded so that for an $(\varepsilon_g, \varepsilon_H)$-SOSP $Y$ of the perturbed problem* (P) *we have:*

$$0 \leq \langle CY, Y \rangle - f^\star \leq (\varepsilon_H + \varepsilon_g^2 \eta)R + \frac{\varepsilon_g}{2}\sqrt{R} + 6\sigma_W\sqrt{n}R,$$

*where $\eta$ is as defined in* (15) *and $\sigma_W^2$ is the variance of the Wigner perturbation $W$.*

# 4 Applications

The approximate global optimality results established in the previous section can be applied to deduce guarantees on the quality of ASOSPs of the low-rank factorization for a number of SDPs that appear in machine learning problems. Of particular interest, we focus on the phase retrieval problem. This problem consists in retrieving a signal $z \in \mathbb{C}^d$ from $n$ amplitude measurements $b = |Az| \in \mathbb{R}_+^n$ (the absolute value of vector $Az$ is taken entry-wise). If we can recover the complex phases of $Az$, then $z$ can be estimated through linear least-squares. Following this approach, Waldspurger et al. [2015] argue that this task can be modeled as the following non-convex problem:

$$
\begin{aligned}
\min_{u \in \mathbb{C}^n} \quad & u^* C u \\
\text{subject to} \quad & |u_i| = 1, \text{ for } i = 1, \dots, n,
\end{aligned}
\tag{PR}
$$

where $C = \mathrm{diag}(b)(I - AA^\dagger)\mathrm{diag}(b)$ and $\mathrm{diag}\colon \mathbb{R}^n \to \mathbb{H}^{n \times n}$ maps a vector to the corresponding diagonal matrix. The classical relaxation is to rewrite the above in terms of $X = uu^*$ (lifting) without enforcing $\mathrm{rank}(X) = 1$, leading to a complex SDP which Waldspurger et al. [2015] call PhaseCut:

$$
\begin{aligned}
\min_{X \in \mathbb{H}^{n \times n}} \quad & \langle C, X \rangle \\
\text{subject to} \quad & \mathrm{diag}(X) = 1, \\
& X \succeq 0.
\end{aligned}
\tag{PhaseCut}
$$

The same SDP relaxation also applies to a problem called angular synchronization [Singer, 2011]. The Burer–Monteiro factorization of (PhaseCut) is an optimization problem over a matrix $Y \in \mathbb{C}^{n \times k}$ as follows:

$$
\begin{aligned}
\min_{Y \in \mathbb{C}^{n \times k}} \quad & \langle CY, Y \rangle \\
\text{subject to} \quad & \mathrm{diag}(YY^*) = 1.
\end{aligned}
\tag{PhaseCut-BM}
$$

For a feasible $Y$, each row has unit norm: the search space is a Cartesian product of spheres in $\mathbb{C}^k$, which is a smooth manifold. We can check that Assumption 1 holds for all $k \geq 1$. Furthermore, the feasible space of the SDP is compact. Therefore, Theorem 3.1 applies.

In this setting, $\mathrm{Tr}(X) = n$ for all feasible $X$, and $\|\mathcal{A}^* \circ G^\dagger \circ \mathcal{A}\|_{\mathrm{op}} = 1$ for all feasible $Y$ (because $G = G(Y) = I_m$ for all feasible $Y$—see Lemma 2.2—and $\mathcal{A}^* \circ \mathcal{A}$ is an orthogonal projector from Hermitian matrices to diagonal matrices). For this reason, the constants defined in (10) and (11) can be set to $R = n$ and $K = 1$.

As a comparison, Mei et al. [2017] also provide an optimality gap for ASOSPs of (PhaseCut) without perturbation. Their result is more general in the sense that it holds for all possible values of $k$.

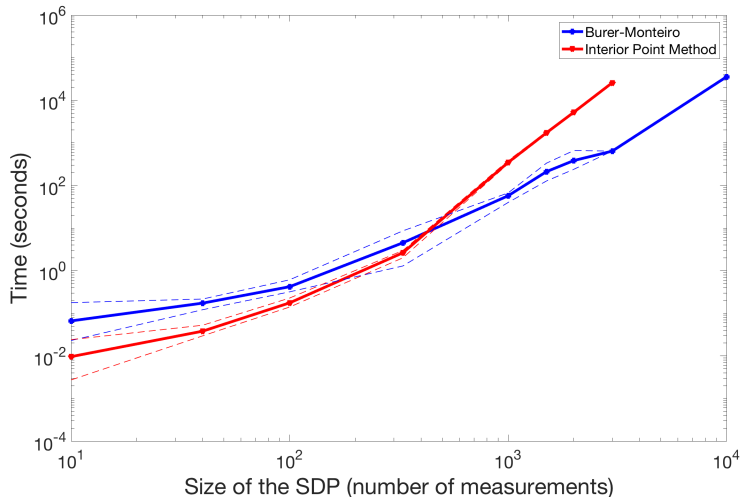

Figure 1: Computation time of the dedicated interior-point method (IPM) and of the Burer–Monteiro approach (BM) to solve (PhaseCut). For increasing values of $n$ (horizontal axis), we display the computation time averaged over four independent realizations of the problem (vertical axis). The smallest and largest observed computation times are represented with dashed lines. At $n = 3000$, BM is about 40 times faster than IPM. For the largest value of $n$, IPM runs out of memory.

However, when $k$ is large, it does not accurately capture the fact that SOSPs are optimal, thus incurring a larger bound on the optimality gap of ASOSPs. In contrast, our bounds do show that for $k$ large enough, as $\varepsilon_g, \varepsilon_H$ go to zero, the optimality gap goes to zero, with the trade-off that they do so for a perturbed problem (though see Corollary 3.1), with high probability.

**Numerical Experiments**

We present the empirical performance of the low-rank approach in the case of (PhaseCut). We compare it with a dedicated interior-point method (IPM) implemented by Helmberg et al. [1996] for real SDPs and adapted to phase retrieval as done by Waldspurger et al. [2015]. This adaptation involves splitting the real and the imaginary parts of the variables in (PhaseCut) and forming an equivalent real SDP with double the dimension. The Burer–Monteiro approach (BM) is implemented in complex form directly using Manopt, a toolbox for optimization on manifolds [Boumal et al., 2014]. In particular, a Riemannian Trust-Region method (RTR) is used [Absil et al., 2007]. Theory supports that these methods can return an ASOSP in a finite number of iterations [Boumal et al., 2018a]. We stress that the SDP is *not* perturbed in these experiments: the role of the perturbation in the analysis is to understand why the low-rank approach is so successful in practice despite the existence of pathological cases. In practice, we do not expect to encounter pathological cases.

Our numerical experiment setup is as follows. We seek to recover a signal of dimension $d$, $z \in \mathbb{C}^d$, from $n$ measurements encoded in the vector $b \in \mathbb{R}_+^n$ such that $b = |Az| + \epsilon$, where $A \in \mathbb{C}^{n \times d}$ is the sensing matrix and $\epsilon \sim \mathcal{N}(0, \mathrm{I}_d)$ is standard Gaussian noise. For the numerical experiments, we generate the vectors $z$ as complex random vectors with i.i.d. standard Gaussian entries, and we randomly generate the complex sensing matrices $A$ also with i.i.d. standard Gaussian entries. We do so for values of $d$ ranging from 10 to 1000, and always for $n = 10d$ (that is, there are 10 magnitude measurements per unknown complex coefficient, which is an oversampling factor of 5.) Lastly, we generate the measurement vectors $b$ as described above and we cap its values from below at 0.01 in order to avoid small (or even negative) magnitude measurements.

For $n$ up to 3000, both methods solve the same problem, and indeed produce the same answer up to small discrepancies. The BM approach is more accurate, at least in satisfying the constraints, and, for $n = 3000$, it is also about 40 times faster than IPM. BM is run with $k = \sqrt{n}$ (rounded up), which is expected to be generically sufficient to include the global optimum of the SDP (as confirmed in practice). For larger values of $n$, the IPM ran into memory issues and we had to abort the process.

## 5 Conclusion

We considered the low-rank (or Burer–Monteiro) approach to solve equality-constrained SDPs. Our key assumptions are that (a) the search space of the SDP is compact, and (b) the search space of its low-rank version is smooth (the actual condition is slightly stronger). Under these assumptions, we proved using smoothed analysis that, provided $k = \tilde{\Omega}(\sqrt{m})$ where $m$ is the number of constraints, if the cost matrix is perturbed randomly, with high probability, approximate second-order stationary points of the perturbed low-rank problem map to approximately optimal solutions of the perturbed SDP. We also related optimality gaps in the perturbed SDP to optimality gaps in the original SDP. Finally, we applied this result to an SDP relaxation of phase retrieval (also applicable to angular synchronization).

## Acknowledgments

NB is partially supported by NSF award DMS-1719558.

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
