[Supplementary Material]

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

## Footnotes

[2]The reference proves the statement for complex matrices with diagonal entries equal to zero. That proof can easily be adapted to the definition of Wigner matrices used in this paper, both real and complex.

[3]The lemma in the reference shows that for any $\varepsilon \in (0,1)$ the cardinality of one such $\varepsilon$-net is bounded by $(3/\varepsilon)^d$. Furthermore, for $\varepsilon \geq 1$, there is an obvious $\varepsilon$-net of cardinality one, comprising just the origin. Hence, for any $\varepsilon > 0$, it is possible so find an $\varepsilon$-net of cardinality at most $\max\left(1, (3/\varepsilon)^d\right) \leq (1 + 3/\varepsilon)^d$.

[4]The same result can be obtained by using Theorem IV.2.14 in the reference. In this setting, one considers the function $F(A) = -\sum_{i=1}^n \sigma_i^2(A)$; then, use the subadditive property of $F$, i.e., $F(A + B) \leq F(A) + F(B)$, and define $A = \bar{M} + W$ and $B = -(M + W)$.

[5]We use that, for any $u, v \geq 0$, $\sqrt{u + v} \leq \sqrt{\sqrt{u}^2 + \sqrt{v}^2 + 2\sqrt{u}\sqrt{v}} = \sqrt{u} + \sqrt{v}$.

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

# A Proof of Lemma 2.2

We follow the proof of [Boumal et al., 2018b, Lemma 2.2] and reproduce it here to be self-contained, and also because the reference treats only the real case; writing the proof here explicitly allows to verify that, indeed, all steps go through for the complex case as well.

Orthogonal projection is along the normal space, so that $\mathrm{Proj}_Y Z$ is in $\mathrm{T}_Y \mathcal{M}$ (3) and $Z - \mathrm{Proj}_Y Z$ is in $\mathrm{N}_Y \mathcal{M}$, where the normal space at $Y$ is (using Assumption 1)

$$\mathrm{N}_Y \mathcal{M} = \left\{ Z \in \mathbb{K}^{n \times k} : \langle Z, \dot{Y} \rangle = 0 \; \forall \dot{Y} \in \mathrm{T}_Y \mathcal{M} \right\} = \mathrm{span}\{A_1 Y, \ldots, A_m Y\}. \tag{16}$$

From the latter we infer there exists $\mu \in \mathbb{R}^m$ such that

$$Z - \mathrm{Proj}_Y Z = \sum_{i=1}^m \mu_i A_i Y = \mathcal{A}^*(\mu) Y,$$

since the adjoint of $\mathcal{A}$ is $\mathcal{A}^*(\mu) = \mu_1 A_1 + \cdots + \mu_m A_m$. Multiply on the right by $Y^*$ and apply $\mathcal{A}$ to obtain

$$\mathcal{A}(ZY^*) = \mathcal{A}(\mathcal{A}^*(\mu)YY^*),$$

where we used $\mathcal{A}(\mathrm{Proj}_Y(Z)Y^*) = 0$ since $\mathrm{Proj}_Y(Z) \in \mathrm{T}_Y \mathcal{M}$. The right-hand side expands into

$$\mathcal{A}(\mathcal{A}^*(\mu)YY^*)_i = \left\langle A_i, \sum_{j=1}^m \mu_j A_j YY^* \right\rangle = \sum_{j=1}^m \langle A_i Y, A_j Y \rangle \, \mu_j = (G\mu)_i,$$

where $G$ is a real, positive semidefinite matrix of size $m$ defined by $G_{ij} = \langle A_i Y, A_j Y \rangle$. By construction, this system of equations in $\mu$ has at least one solution; we single out $\mu = G^\dagger \mathcal{A}(ZY^*)$, where $G^\dagger$ is the Moore–Penrose pseudo-inverse of $G$. The function $Y \mapsto G^\dagger$ is continuous and differentiable at $Y \in \mathcal{M}$ provided $G$ has constant rank in an open neighborhood of $Y$ in $\mathbb{K}^{n \times k}$ [Golub and Pereyra, 1973, Thm 4.3], which is the case for all $Y \in \mathcal{M}$ under Assumption 1.

# B Lower-bound for smallest singular values

This appendix provides supporting results necessary for Appendix C, which is devoted to the proof of Lemma 3.1. The statements we need are established for $\mathbb{K} = \mathbb{R}$ in [Bhojanapalli et al., 2018, Cor. 5, Lem. 7]. Here we give the corresponding statements for $\mathbb{K} = \mathbb{C}$: the proofs are essentially the same.

We first state a special case of Corollary 1.17 from [Nguyen, 2018]. Here, $N_I(X)$ denotes the number of eigenvalues of $X \in \mathbb{S}^{n \times n}$ in the real interval $I$. (Note that the reference covers the real case in its main statement, and addresses the complex case later on as a remark.) For Gaussian Wigner matrices, we follow Definition 3.1. Furthermore, $\mathbb{P}\{E\}$ denotes the probability of event $E$.

**Corollary B.1.** *Let $\overline{M}$ be a deterministic Hermitian matrix of size $n$. Let $\overline{W}$ be a Gaussian Wigner matrix with variance 1. Then, for any given $0 < \gamma < 1$, there exists a constant $c = c(\gamma)$ such that for any $\varepsilon > 0$ and $k \geq 1$, with $I$ being the interval $\left[ -\dfrac{\varepsilon k}{\sqrt{n}}, \dfrac{\varepsilon k}{\sqrt{n}} \right]$,*

$$\mathbb{P}\left\{ N_I(\overline{M} + \overline{W}) \geq k \right\} \leq n^k \left( \frac{c\varepsilon}{\sqrt{2\pi}} \right)^{(1-\gamma)k^2/2}.$$

The next lemma follows easily—the original proof for $\mathbb{K} = \mathbb{R}$ is in [Bhojanapalli et al., 2018, Lem. 7].

**Lemma B.1.** *Let $M$ be a deterministic Hermitian matrix of size $n$. Let $W$ be a complex Gaussian Wigner matrix of size $n$ with variance $\sigma_W^2$, independent of $M$. There exists an absolute constant $c_0$ such that:*

$$\mathbb{P}\left\{ \sum_{i=1}^k \sigma_{n-(i-1)}(M+W)^2 < \frac{k^2 \sigma_W^2}{c_0 n} \right\} \leq \exp\left( -\frac{k^2}{8} \log(8\pi) + k \log(n) \right).$$

*Proof.* In our case, the entries of $W$ have variance $\mathbb{E}[|W_{i,j}|^2] = \sigma_W^2$. Thus, set $W = \sigma_W \overline{W}$ and $M = \sigma_W \overline{M}$. From Corollary B.1, we get

$$N_{\sigma_W I}(M + W) = N_I(\overline{M} + \overline{W}) < k$$

with probability at least $1 - n^k \left(\dfrac{c\varepsilon}{\sqrt{2\pi}}\right)^{(1-\gamma)k^2/2}$. In this event, $\sigma_{n-(k-1)}(\overline{M} + \overline{W}) \geq \dfrac{\varepsilon k}{\sqrt{n}} \sigma_W$.

With the choices $\gamma = \dfrac{1}{2}$ and $\varepsilon = \dfrac{1}{2c}$, we get that

$$\sigma_{n-(k-1)}(\overline{M} + \overline{W}) \geq \frac{k}{2c\sqrt{n}} \sigma_W$$

with probability at least $1 - \exp(-\frac{k^2}{8}\log(8\pi) + k\log(n))$. In that event,

$$\sum_{i=1}^{k} \sigma_{n-(i-1)}(\overline{M} + \overline{W})^2 \geq \sigma_{n-(k-1)}(\overline{M} + \overline{W})^2 \geq \frac{k^2}{c_0 n} \sigma_W^2$$

for some absolute constant $c_0 = 4c^2$. $\qquad\square$

## C Proof of Lemma 3.1

This section builds on a result from [Bhojanapalli et al., 2018], where a similar statement was made under different assumptions. The proof follows closely the developments therein, with appropriate changes. Using (8), $Y$ is an $\varepsilon_g$-FOSP of the perturbed problem if and only if $\|2SY\| \leq \varepsilon_g$ with $S = M + W$ and

$$M = C - (\mathcal{A}^* \circ G^\dagger \circ \mathcal{A})((C + W)YY^*). \tag{17}$$

Let $Y = P\Sigma Q^*$ be a thin SVD of $Y$, where $P$ is $n \times k$ with orthonormal columns (assuming without loss of generality $k \leq n$, as otherwise $\sigma_k(Y) = 0$ deterministically) and $Q$ is $k \times k$ orthogonal. Then,

$$\begin{aligned}
\varepsilon_g \geq \|2SY\| &= \|2(M + W)Y\| \\
&\geq 2\sigma_k(Y)\|(M + W)P\| \\
&\geq 2\sigma_k(Y)\sqrt{\sum_{i=1}^{k} \sigma_{n-(i-1)}(M + W)^2}.
\end{aligned}$$

Thus, we control the smallest singular value of $Y$ in terms of $\varepsilon$ and the $k$ smallest singular values of $M + W$:

$$\sigma_k(Y) \leq \frac{\varepsilon_g}{2\sqrt{\sum_{i=1}^{k} \sigma_{n-(i-1)}(M + W)^2}}. \tag{18}$$

Given that $M$ is not statistically independent of $W$, we are not able to directly apply Lemma B.1. Indeed, $M$ depends on $W$ and on $Y$, and $Y$ itself is an $\varepsilon_g$-FOSP of the perturbed problem: a feature which depends on $W$. To tackle this issue, we cover the set of possible $M$s with a net. Lemma B.1 provides a bound for each $\overline{M}$ in this net. By union bound, we can extend the lemma for all $\overline{M}$. By taking a sufficiently dense net, we then infer that $M$ is necessarily close to one of these $\overline{M}$'s and conclude.

To this end, we first control $\|M - C\|$ using the definitions of $R$ (10) and $K$ (11):

$$\begin{aligned}
\|M - C\| &= \|\mathcal{A}^* \circ G^\dagger \circ \mathcal{A}((C + W)YY^*)\| \\
&\leq \|\mathcal{A}^* \circ G^\dagger \circ \mathcal{A}\|_{\mathrm{op}}\|C + W\|_{\mathrm{op}}\|YY^*\| \\
&\leq K(\|C\|_{\mathrm{op}} + \|W\|_{\mathrm{op}})R.
\end{aligned}$$

Since $W$ is a Gaussian Wigner matrix with variance $\sigma_W^2$, it is a well known fact (see for instance[2] Part 1 of Appendix A in [Bandeira et al., 2017]) that, with probability at least $1 - e^{-\frac{n}{2}}$,

$$\|W\|_{\mathrm{op}} \leq 3\sigma_W\sqrt{n}. \tag{19}$$

Hence, with probability at least $1 - e^{-\frac{n}{2}}$,

$$\|M - C\| \leq RK(\|C\|_{\mathrm{op}} + 3\sigma_W\sqrt{n}) \triangleq \kappa,$$

where we recover $\kappa$ as defined in (12).

As a result, $M$ lies in a ball of center $C$ and radius $\kappa$. Moreover, from (17), we remark that $M$ lives in an affine subspace of dimension $\mathrm{rank}(\mathcal{A}^* \circ G^\dagger \circ \mathcal{A}) = \mathrm{rank}(\mathcal{A})$. A unit ball in Frobenius norm in $d$ dimensions admits an $\varepsilon$-net of $\left(1 + \frac{3}{\varepsilon}\right)^d$ points (see for instance Lemma 1.18 in [Rigollet and Hütter, 2017]).[3] Thus, we pick a $\frac{k\sigma_W}{2\kappa\sqrt{c_0 n}}$-net on the unit ball with $\left(1 + \frac{6\kappa\sqrt{c_0 n}}{k\sigma_W}\right)^{\mathrm{rank}(\mathcal{A})}$ points. Rescaling by a factor $\kappa$ gives a $\frac{k\sigma_W}{2\sqrt{c_0 n}}$-net of a ball of radius $\kappa$ centered at zero. Hence, for any $M$ as in (17) there necessarily exists a point $\bar{M}$ in the net satisfying:

$$\|\bar{M} - M\| \leq \frac{k\sigma_W}{2\sqrt{c_0 n}}. \tag{20}$$

Let $T\colon \mathbb{S}^{n\times n} \to \mathbb{R}^k$ be defined by $T_q(A) = (\sigma_{n-q+1}(A), \ldots, \sigma_n(A))^\top$, that is: $T$ extracts the $q$ smallest singular values of $A$, in order. Then, by using the result from Exercise IV.3.5. in [Bhatia, 2007] in the first inequality,[4] we have:

$$\begin{aligned}
\|\bar{M} - M\| &= \|(\bar{M} + W) - (M + W)\| \\
&= \sqrt{\sum_{i=1}^n \sigma_i^2\left((\bar{M} + W) - (M + W)\right)} \\
&\geq \sqrt{\sum_{i=1}^n \left(\sigma_i(\bar{M} + W) - \sigma_i(M + W)\right)^2} \\
&= \|T_n(\bar{M} + W) - T_n(M + W)\| \\
&\geq \|T_k(\bar{M} + W) - T_k(M + W)\| \\
&\geq \|T_k(\bar{M} + W)\| - \|T_k(M + W)\|,
\end{aligned}$$

where we used the triangular inequality in the last inequality. Thus, rearranging we obtain

$$\sqrt{\sum_{i=1}^k \sigma_{n-(i-1)}(M + W)^2} \geq \sqrt{\sum_{i=1}^k \sigma_{n-(i-1)}(\bar{M} + W)^2} - \|\bar{M} - M\|. \tag{21}$$

Taking a union bound for Lemma B.1 over each $\bar{M}$ in the net, we get that

$$\sqrt{\sum_{i=1}^k \sigma_{n-(i-1)}(\bar{M} + W)^2} \geq \frac{k\sigma_W}{\sqrt{c_0 n}} \tag{22}$$

holds with probability at least

$$1 - \exp\left(-\frac{k^2}{8}\log(8\pi) + k\log(n) + \mathrm{rank}(\mathcal{A}) \cdot \log\left(1 + \frac{6\kappa\sqrt{c_0 n}}{k\sigma_W}\right)\right). \tag{23}$$

Combining (20), (21) and (22), we conclude that

$$\sqrt{\sum_{i=1}^k \sigma_{n-(i-1)}(M + W)^2} \geq \frac{k\sigma_W}{2\sqrt{c_0 n}} \tag{24}$$

holds with probability bounded as in (23). Combining with (18), we obtain

$$\sigma_k(Y) \leq \frac{\varepsilon_g}{\sigma_W} \frac{\sqrt{c_0 n}}{k}$$

as desired. It remains to discuss the probability of success, which we do below.

Inside the log in (23), we can safely replace $k$ with 1, as this only hurts the probability. Then, the result holds with probability at least

$$1 - \exp\left(-\frac{k^2}{8}\log(8\pi) + k\log(n) + \mathrm{rank}(\mathcal{A}) \cdot \log\left(1 + \frac{6\kappa\sqrt{c_0 n}}{\sigma_W}\right)\right).$$

We would like to constrain $k$ such that the exponential part is bounded by $\delta$. In this fashion, taking a union bound with event (19), we will get an overall probability of success of at least $1 - \delta - e^{-\frac{n}{2}}$. Equivalently, $k$ must satisfy the quadratic inequality

$$-ak^2 + bk + c \leq \log(\delta),$$

with $a, b > 0$, $c \geq 0$ defined by $a = \frac{\log(8\pi)}{8}, b = \log(n), c = \mathrm{rank}(\mathcal{A}) \cdot \log\left(1 + \frac{6\kappa\sqrt{c_0 n}}{\sigma_W}\right)$. This quadratic inequality can be rewritten as:

$$ak^2 - bk - c' \geq 0,$$

with $c' = c + \log(1/\delta)$. This quadratic has two distinct real roots, one positive and one negative:

$$\frac{b \pm \sqrt{b^2 + 4ac'}}{2a}.$$

Since $k$ is positive, we deduce that $k$ needs to be larger than the positive root. The latter obeys the following inequality:[5]

$$\frac{b + \sqrt{b^2 + 4ac'}}{2a} \leq \frac{b + b + 2\sqrt{ac'}}{2a} = \frac{b + \sqrt{ac'}}{a} = \frac{1}{a}b + \frac{1}{\sqrt{a}}\sqrt{c'}.$$

Since both $1/a$ and $1/\sqrt{a}$ are smaller than 3, it is sufficient to require

$$k \geq 3\left(b + \sqrt{c + \log(1/\delta)}\right).$$

Assuming $\delta \leq 1$, we can use the inequality in the footnote again and find that it is sufficient to have

$$k \geq 3\left(b + \sqrt{\log(1/\delta)} + \sqrt{c}\right).$$

Plugging in the definitions of $b$ and $c$, we find the sufficient condition (with $\delta \leq 1$):

$$k \geq 3\left[\log(n) + \sqrt{\log(1/\delta)} + \sqrt{\mathrm{rank}(\mathcal{A}) \cdot \log\left(1 + \frac{6\kappa\sqrt{c_0 n}}{\sigma_W}\right)}\right].$$

Since $\mathrm{rank}(\mathcal{A}) \leq m$, we obtain the desired sufficient bound on $k$.

## D  Proof of Lemma 3.2

The Riemannian gradient and Hessian of the objective function $g$ of (P) are respectively given by equations (8) and (9). Since $Y$ is an $(\varepsilon_g, \varepsilon_H)$-SOSP, it holds for all $\dot{Y} \in \mathrm{T}_Y \mathcal{M}$ (3) with $\|\dot{Y}\| = 1$ that:

$$-\varepsilon_H \leq \frac{1}{2}\left\langle \dot{Y}, \mathrm{Hess}\, g(Y)[\dot{Y}]\right\rangle = \langle \dot{Y}, S\dot{Y}\rangle. \tag{25}$$

Our goal is to show that $S$ is almost positive semidefinite. To this end, we first construct specific $\dot{Y}$'s to exploit the fact that $Y$ is almost rank deficient. Let $z \in \mathbb{K}^k$ be a right singular vector of $Y$ such that $\|Yz\| = \sigma_k(Y)$ and $\|z\| = 1$. For any $x \in \mathbb{K}^n$ with $\|x\| = 1$, we introduce $U = xz^*$. Decompose $U$

in two components: $U = U_T + U_{T\perp}$, with $U_T$ the component of $U$ in the tangent space $\mathrm{T}_Y\mathcal{M}$ and $U_{T\perp}$ the orthogonal component in $\mathrm{N}_Y\mathcal{M}$. Given that $\|z\| = 1$, using (25) with $\dot{Y} = U_T$, we have:

$$
\begin{aligned}
\langle x, Sx \rangle = \langle U, SU \rangle &= \langle U_T, SU_T \rangle + 2\langle U_{T\perp}, SU_T \rangle + \langle U_{T\perp}, SU_{T\perp} \rangle \\
&\geq -\varepsilon_H \|U_T\|^2 + 2\langle U_{T\perp}, SU_T \rangle + \langle U_{T\perp}, SU_{T\perp} \rangle \\
&\geq -\varepsilon_H + 2\langle U_{T\perp}, SU_T \rangle + \langle U_{T\perp}, SU_{T\perp} \rangle \\
&= -\varepsilon_H + 2\langle U_{T\perp}, SU \rangle - \langle U_{T\perp}, SU_{T\perp} \rangle,
\end{aligned}
\tag{26}
$$

where we also used $\|U_T\|^2 \leq \|U\|^2 = 1$. We know by Lemma 2.2 that $U_T$ can be written as:

$$
U_T = \mathrm{Proj}_Y U = xz^* - \mathcal{A}^* \left( G^\dagger \mathcal{A}\left(xz^*Y^*\right) \right) Y.
\tag{27}
$$

Therefore, the component along the normal space, $U_{T\perp}$, is:

$$
U_{T\perp} = \mathcal{A}^* \left( G^\dagger \mathcal{A}\left(xz^*Y^*\right) \right) Y.
\tag{28}
$$

Using (28), we can derive an upper bound on $\langle U_{T\perp}, SU_{T\perp} \rangle$. Indeed, by Cauchy–Schwarz we obtain:

$$
\langle U_{T\perp}, SU_{T\perp} \rangle \leq \|U_{T\perp}\|^2 \|S\|_{\mathrm{op}}.
$$

From the expression for $S$ in (7) and the definitions of $R$ (10) and $K$ (11), the two factors are easily bounded since $\|xz^*Y^*\| = \|Yz\| = \sigma_k(Y)$:

$$
\|U_{T\perp}\| \leq \|\mathcal{A}^* \circ G^\dagger \circ \mathcal{A}\|_{\mathrm{op}} \|xz^*Y^*\| \|Y\| \leq K\sqrt{R} \cdot \sigma_k(Y),
$$

and

$$
\begin{aligned}
\|S\|_{\mathrm{op}} &\leq \|C\|_{\mathrm{op}} + \|\mathcal{A}^*(G^\dagger\mathcal{A}(CYY^*))\|_{\mathrm{op}} \\
&\leq \|C\|_{\mathrm{op}} + \|\mathcal{A}^* \circ G^\dagger \circ \mathcal{A}\|_{\mathrm{op}} \|CYY^*\| \leq (1 + KR)\|C\|_{\mathrm{op}}.
\end{aligned}
$$

Combining, we find the bound

$$
\langle U_{T\perp}, SU_{T\perp} \rangle \leq K^2 R(1 + KR)\|C\|_{\mathrm{op}} \cdot \sigma_k(Y)^2.
\tag{29}
$$

Through a similar reasoning, we can handle the remaining term in (26). The important step is to make sure $\sigma_k(Y)$ appears quadratically:

$$
\begin{aligned}
\langle U_{T\perp}, SU \rangle &= \left\langle \mathcal{A}^*\left(G^\dagger\mathcal{A}\left(xz^*Y^*\right)\right)Y, Sxz^* \right\rangle \\
&= \left\langle (\mathcal{A}^* \circ G^\dagger \circ \mathcal{A})(xz^*Y^*), Sxz^*Y^* \right\rangle \\
&\geq -\|\mathcal{A}^* \circ G^\dagger \circ \mathcal{A}\|_{\mathrm{op}} \|S\|_{\mathrm{op}} \|xz^*Y^*\|^2 \\
&\geq -K(1 + KR)\|C\|_{\mathrm{op}} \cdot \sigma_k(Y)^2.
\end{aligned}
\tag{30}
$$

Finally, combining (29) and (30) with (26) yields:

$$
\begin{aligned}
\langle x, Sx \rangle &\geq -\varepsilon_H - 2K(1 + KR)\|C\|_{\mathrm{op}} \cdot \sigma_k(Y)^2 - K^2 R(1 + KR)\|C\|_{\mathrm{op}} \cdot \sigma_k(Y)^2 \\
&= -\varepsilon_H - K(2 + KR)(1 + KR)\|C\|_{\mathrm{op}} \cdot \sigma_k(Y)^2 \\
&\geq -\varepsilon_H - K(2 + KR)^2\|C\|_{\mathrm{op}} \cdot \sigma_k(Y)^2 \\
&= -\varepsilon_H - \zeta\|C\|_{\mathrm{op}} \cdot \sigma_k(Y)^2,
\end{aligned}
$$

where $\zeta$ is as defined in the lemma statement. This holds for any unit vector $x$, hence the proof is complete.

## E   Proof of Theorem 3.1

We now build on Lemmas 3.1 and 3.2 to prove Theorem 3.1. The first part of the argument is fully deterministic: it relates the minimal eigenvalue of $S$ to the optimality gap of the optimization problem.

Let $Y$ be an $(\varepsilon_g, \varepsilon_H)$-SOSP of problem (P) with perturbed cost matrix $\tilde{C} = C + W$. By Lemma 3.2 applied to the perturbed problem,

$$
\lambda_{\min}(S) \geq -\varepsilon_H - \zeta\|\tilde{C}\|_{\mathrm{op}}\sigma_k(Y)^2,
\tag{31}
$$

where $\zeta$ is as defined in that lemma, and $S$ is as defined in (7) with cost matrix $\tilde{C}$ instead of $C$:

$$S(Y) = \tilde{C} - \mathcal{A}^*(\mu(Y)), \text{ and}$$
$$\mu(Y) = G^\dagger \mathcal{A}(\tilde{C}YY^*).$$

Using the definition of $\mathcal{C}$, for all $X' \in \mathcal{C}$ feasible for the problem (SDP),

$$\lambda_{\min}(S) \cdot \operatorname{Tr}(X') \le \langle S(Y), X' \rangle = \langle \tilde{C}, X' \rangle - \langle \mathcal{A}^*(\mu(Y)), X' \rangle = \langle \tilde{C}, X' \rangle - \langle \mu(Y), b \rangle.$$

In particular

$$\langle \mu(Y), b \rangle = \langle \mu(Y), \mathcal{A}(YY^*) \rangle = \langle \tilde{C} - S(Y), YY^* \rangle = g(Y) - \langle S(Y)Y, Y \rangle.$$

Combining those equations, using $\operatorname{grad} g(Y) = 2S(Y)Y$ and taking $X' = X^*$, we find

$$0 \le g(Y) - f^\star \le -\lambda_{\min}(S) \cdot \operatorname{Tr}(X^*) + \frac{1}{2} \langle \operatorname{grad} g(Y), Y \rangle$$
$$\le -\lambda_{\min}(S) \cdot \operatorname{Tr}(X^*) + \frac{\varepsilon_g}{2} \|Y\|.$$

Since $\mathcal{C}$ is compact, we use the definition of $R$ in (10) to get that $\operatorname{Tr}(X^*) \le R$ and $\|Y\| \le \sqrt{R}$:

$$0 \le g(Y) - f^\star \le -\lambda_{\min}(S) \cdot R + \frac{\varepsilon_g}{2}\sqrt{R}$$
$$\le \left( \varepsilon_H + \zeta \|\tilde{C}\|_{\mathrm{op}} \sigma_k(Y)^2 \right) R + \frac{\varepsilon_g}{2}\sqrt{R}, \tag{32}$$

where we used (31) in the last step.

We can now turn to the probabilistic part of the proof. Using Lemma 3.1, we have with probability at least $1 - \delta - e^{-\frac{n}{2}}$ that

$$\|W\|_{\mathrm{op}} \le 3\sigma_W \sqrt{n}, \text{ and}$$
$$\sigma_k(Y) \le \frac{\varepsilon_g}{\sigma_W} \frac{\sqrt{c_0 n}}{k},$$

and, by assumption,

$$k \ge 3 \left[ \log(n) + \sqrt{\log(1/\delta)} + \sqrt{m \cdot \log\left(1 + \frac{6\kappa\sqrt{c_0 n}}{\sigma_W}\right)} \right] \ge 3\sqrt{m \cdot \log\left(1 + \frac{6\kappa\sqrt{c_0 n}}{\sigma_W}\right)}.$$

In that event, combining, it follows that

$$\|\tilde{C}\|_{\mathrm{op}} \le \|C\|_{\mathrm{op}} + 3\sigma_W \sqrt{n}, \text{ and}$$
$$\sigma_k(Y)^2 \le \varepsilon_g^2 \frac{c_0 n}{9m\sigma_W^2 \log\left(1 + \frac{6\kappa\sqrt{c_0 n}}{\sigma_W}\right)}.$$

Combining with the deterministic result (32), we find that the optimality gap is bounded as

$$0 \le g(Y) - f^\star \le \left( \varepsilon_H + \varepsilon_g^2 \eta \right) R + \frac{\varepsilon_g}{2}\sqrt{R},$$

where $\eta$ is as defined in (15). This concludes the proof.

# F   Proof of Corollary 3.1

Consider the two following functions:

$$f(C) = \min_{X \in \mathcal{C}} \langle C, X \rangle, \qquad\qquad h(C) = \max_{X \in \mathcal{C}} \langle C, X \rangle.$$

By assumption on $X$,

$$\langle C, X \rangle - \langle -W, X \rangle = \langle (C + W), X \rangle \le f(C + W) + \varepsilon_f.$$

We can rearrange and get:

$$\langle C, X \rangle \leq f(C + W) + \varepsilon_f + \langle -W, X \rangle \leq f(C + W) + \varepsilon_f + h(-W).$$

Moreover,

$$f(C + W) = \min_{X \in \mathcal{C}} \left( \langle C, X \rangle + \langle W, X \rangle \right) \leq f(C) + h(W).$$

Overall, we get a bound on the optimality gap, using that $f(C) = f^\star$:

$$\langle C, X \rangle - f^\star \leq \varepsilon_f + h(W) + h(-W).$$

To conclude, observe that

$$h(W) = \max_{X \in \mathcal{C}} \langle W, X \rangle \leq \|W\|_{\mathrm{op}} \max_{X \in \mathcal{C}} \mathrm{Tr}(X) \leq \|W\|_{\mathrm{op}} R,$$

where we used that $\mathrm{Tr}(X) \leq R$ for all $X \in \mathcal{C}$. The same bound applies to $h(-W)$.