[Reviews · NeurIPS 2018]

Reviewer 1



I have read the author response. Thank you. This paper improves upon existing works by showing global optimality of approx second order stationary points under mild assumptions. This is important in theory and practice both, since finite precision allows only approximate solutions. A study for the special case of phase cut is also presented. The paper is very notation heavy but clearly written and easy to follow. I have gone through the main proofs. The empirical section is insignificant but this is mainly a theory paper, so I'll vote for acceptance.

Reviewer 2



This paper studies the connection between an SDP and its nonconvex QP equivalent based on Burer-Monteiro low-rank factorization. The authors analyzed how an approximate SOSP of the QP reformulation approximates the optimal solution of the original SDP. The results are important for our understanding of the Burer-Monteiro approach. In general, I found this is a good paper. One possible drawback is that the authors didn't discuss how to find an approximate SOSP for the QP. Moreover, in the numerical tests, it is better to show how the approximate SOSP is obtained and whether it is close to the optimal solution to the SDP or not.

Reviewer 3



This paper concerns the Burer-Monteiro (i.e. low-rank) factorization heuristic for solving SDP problems of the form min s.t. A(X) = b, X \succeq 0. The BM heuristic relies on the fact when the constraint set is compact, the above SDP has a global solution with rank up to square root of the **number of the linear constraints** (denoted as m). This allows one to perform substitution X = YY^* and then perform a heuristic numerical search in the low-dimensional subspace of Y. In general, there is no guarantee such heuristic search would find the global solution, i.e. Y such that YY^* solves the original SDP. Recent advances (Bounal et al. 2016b, 2018) show that for almost all matrix C, when A(YY^*) defines a smooth manifold, all second-order stationary points (SOSP) of the resulting nonconvex program are global optimal, provided the rank of Y scales as sqrt(m). In practice, numerical methods can at least find approximate SOSP in finite running time. This paper shows that on average (due to the smoothed analysis), all approximate SOSP's produce approximate global optima to the original SDP (measured in terms of the objective value), again so long as the rank of Y scales as sqrt(m). Solving large-scale SDP is of broad interest to many scientific communities with computational components, including machine learning. The current work, combined with the other recent works, build a solid theoretical foundation for the practical relevance and viability of the BM heuristic. The result is closely connected to another recent work Bhojanapalli et al 2018. The two papers address the same stability issue and share some similar points at the technical level. One importance difference is that the current work focuses on a feasible (constrained) version, whereas the other work deals with the penalty version in which the feasibility may not be guaranteed. They work under different technical conditions also, e.g., whether or not A(YY^*) defines a smooth manifold. Additional comments: * Theorem 1.1: From hindsight, it is clear that the perturbation is to avoid the worst cases so that a result can be stated for all C. But around theorem 1.1, this is not clearly explained. This could be confusing at an initial read. * Eq (5): After the first (, I think it should be C instead of CY. * Fig 1: What do you mean by size of the instance for the horizontal axis? It seems confusing if one were to read it against the last paragraph of the numerical experiment section. * For the phase retrieval experiment, curious to know if the solver always returns approximate rank-one solutions.